# Evaluation of influencing factors of China university teaching quality based on fuzzy logic and deep learning technology

Jie Yu 🔟 *

College of Automation Engineering, Shanghai University of Electric Power, Shanghai, China

* yujie_vip@outlook.com

**Citation:** Yu J (2024) Evaluation of influencing factors of China university teaching quality based on fuzzy logic and deep learning technology. PLoS ONE 19(9): e0303613. https://doi.org/10.1371/journal.pone.0303613

**Data Availability Statement:** All relevant data are within the paper.

**Funding:** The author(s) received no specific funding for this work.

## Abstract

Nowadays, colleges and universities focus on the assessment model for considering educational offers, suitable environments, and circumstances for students' growth, as well as the influence of Teaching Quality (TQ) and the applicability of the skills promoted by teaching to life. Teaching excellence is an important evaluation metric at the university level, but it is challenging to determine it accurately due to its wide range of influencing factors. Fuzzy and Deep Learning (DL) approaches must be could to build an assessment model that can precisely measure the teaching qualities to enhance accuracy. Combining fuzzy logic and DL can provide a powerful approach for assessing the influencing factors of college and university teaching effects by implementing the Sequential Intuitionistic Fuzzy (SIF) assisted Long Short-Term Memory (LSTM) model proposed. Sequential Intuitionistic Fuzzy (SIF) can be used sets to assess factors that affect teaching quality to enhance teaching methods and raise the standard of education. LSTM model to create a predictive model that can pinpoint the primary factors that influence teaching quality and forecast outcomes in the future using those influencing factors for academic growth. The enhancement of the SIF-LSTM model for assessing the influencing factors of teaching quality is proved by the accuracy of 98.4%, Mean Square Error (MSE) of 0.028%, Tucker Lewis Index (TLI) measure for all influencing factors and entropy measure of non-membership and membership degree correlation of factors related to quality in teaching by various dimensional measures. The effectiveness of the proposed model is validated by implementing data sources with a set of 60+ teachers' and students' open-ended questionnaire surveys from a university.

## 1. Introduction

Assessment of teaching quality [1] is a challenging, unpredictable system issue influenced by numerous variables such as class size, learning environment, teacher-student relationship, learning outcomes etc. Developing quality assessments aims to build an appropriate connection between the teaching quality assessment indicator and the teaching effect. Universities offer educational resources, classroom interaction, timetables, teaching methodology, and other factors influencing teaching. The basis of a university is teaching, and education in the

**Competing interests:** The authors have declared that no competing interests exist.

classroom is the most essential, central, and integral aspect of teaching and the primary setting for staff development. A fair assessment of the quality of instruction in the classroom is a strong assurance that both teaching quality and management grades will be improved.

A mathematical method known as fuzzy logic addresses data imprecision and uncertainty. It allows for modelling vague and ambiguous concepts by assigning degrees of membership to different categories. This approach can be applied to education by identifying and quantifying the factors that impact teaching effects. By using fuzzy logic to identify the factors that impact teaching effectiveness and then analyzing large datasets of student performance data in different contexts, DL algorithms can identify the factors most closely associated with improved learning outcomes. On the other hand, deep learning is a machine learning type that uses neural networks to learn patterns and relationships for analyzing large datasets of student performance data and identify patterns and relationships that may impact teaching effectiveness.

To improve teaching quality, colleges and universities must consider the influencing factors like characteristics and psychological behaviour of teachers more while developing novel quality initiatives for educational improvement [1]. For more teachers' demands, they may select between classroom teaching and online interactive teaching, which may also be utilized to choose academic effectiveness assessment measures [2]. Universities with various fundamental curriculum requirements are encouraged to use the binary relative assessment approach to analyze the influencing factors that affect the efficacy of teaching quality in experiment design across every university [3]. The assessment of teachers cannot only do qualitative and quantitative scrutiny, but it can also give teachers constructive criticism for further improvement. Focused upgrades are developed to encourage a higher education teaching level and quality [4]. Recognizes the factors that influence the success of instruction in education through efficient and intelligent approaches, such as utilizing technologies like the WEKA tool for problem-solving. This is especially pertinent in the context of the changing educational environment, where Artificial Intelligence technologies are vital for increasing student learning experiences and overall educational advancement. [5]. AI in college education offers a wide range of opportunities to benefit teachers and students more, simplify the teaching-learning process, and elevate the standing of educational institutions. AI makes algorithms learn on their own by utilizing disciplines like Machine Learning (ML), DL, and Natural Language Processing (NLP) [6]. The effectiveness of assessing college teachers' English instruction can be effectively increased by combining Fuzzy Rules (FR) with Neural Networks (NN) approaches. Analytical procedures found in fuzzy rules provide a useful way to assess the several aspects that affect the quality of instruction in higher education. However, conducting in-depth observations that connect with critical elements of teaching quality is necessary in order to fully address these challenges [7]. Classifying and enhancing gathered collection based on knowledge semantics and constraints using fuzzy-optimized data management (FDM). Information dependencies are assessed following the relationships developed between the data based on the value of attributes [8]. An influence factors analysis model was established to identify the effect of teaching in the dance domain and the depth of its consequences using a fuzzy data analysis-based decision tree model [9]. The creation of an influence factors analysis model is frequently an iterative procedure, where the model is continuously improved as further data is obtained or as the comprehension of the phenomena grows more profound. The objective is to develop an effective tool for thoroughly analyzing the numerous aspects that contribute to the observed results. Teaching resources should incorporate the fuzzy clustering technique to assess the divergence between data points and the centroid of the grid, particularly in scenarios with vast datasets. Moreover, these resources should aim to ascertain the accuracy of teaching effectiveness and detect any instances of missing data [10]. A fuzzy comprehensive evaluation model based on Intuitive Fuzzy Information Theory is utilized for handling the challenge of value

orientation teaching in college English that directly affects the students' English proficiency and identifies the relevant factors that influence only a few fundamental teaching processes [11]. A system that assesses teaching quality was developed, an in-depth assessment method combining hierarchical inspection and multi-attribute fuzzy decision-making was proposed, and its applicability in teaching the tourism management speciality was confirmed [12]. A broad assessment of college teacher teaching abilities with the capacity of college teachers has been assessed using the technical method of fuzzy structure element employing IF multi-attribute based decision-making method, and its viability was confirmed [13]. DL is the standard for students' autonomous learning effects and is an emerging expectation for educational institutions teaching today. It backs both rigorous curriculum creation and teaching improvement and the advancement of students' complete comprehension of the material they have learned. Improving educational results and guaranteeing student success necessitates assessing the factors that affect how teaching works in colleges and universities. The influence of teaching quality, which includes various aspects such as interactions between instructors and students, course material, and student participation, makes this endeavour difficult. Conducted a thorough literature study to identify the existing research and theories related to teaching effectiveness, fuzzy logic, and deep learning techniques implementation in determining the elements influencing college teaching effectiveness. However, the theory of fuzzy sets only allows for introducing a single membership functioning, leaving other kinds of uncertainty unaccounted for. Thus, the non-membership and hesitant function introduced by the intuitionistic fuzzy information set theory is implemented with DL techniques. The motivation to understand these intricate relationships would provide information on additional important aspects of teaching quality, which would help colleges and universities develop their response procedures. It will help develop a research framework and identify gaps that can be addressed. Fuzzy and deep learning algorithms can be used to assess the TQ in higher education more precisely and effectively. Fuzzy logic describes and interprets ambiguous or inaccurate information in this combination, while deep learning techniques are used to learn and classify influencing factors of teaching quality. The Sequential Intuitionistic Fuzzy (SIF) aided Long Short-Term Memory (LSTM) model is a hybrid approach for assessing the factors influencing the teaching quality in colleges that incorporates the strengths of both LSTM networks and intuitionistic fuzzy sets.

The findings of this study make a major contribution to the research at this time since they shed light on the general organization of the factors influencing the quality of teaching in colleges and universities. Incorporating the benefits of both LSTM networks and IF sets, the SIF-LSTM model is a hybrid approach that assesses the factors that affect teaching quality among colleges and universities. The proposed model SIF-LSTM determines the key factors that lead to effective teaching outcomes and offers suggestions for how these aspects might be enhanced.

The main contribution of this paper is to investigate the influencing factors of teaching quality assessment and propose an assessment model to allow

1. The IF set is employed to model uncertainties and imprecision in influencing factors of teaching quality, including classroom management, teaching strategies, student participation, and teacher experience. By incorporating membership functions of 3 states, the degree of correlation among factors with teaching quality is identified using an evaluated fuzzy score.

2. Followed by incorporating a DL model called LSTM, the input categories of influencing aspects are given into hidden layers to categorize the length of factors affecting teaching

quality. The assessment model helps decision-makers of universities to allocate assets, aggregate staffing resources, and develop initiatives for teaching.

3. The proposed model is validated by evaluating the metric of fuzzy entropy, accuracy, TLI measure, and MSE values for enhancing the teaching quality assessment model in higher education.

The remaining of the manuscript is arranged in the following manner. Section 2 details the related work for evaluating university teaching quality by various techniques. Section 3 discusses the implementation of sequential-based IF information theory with a judgment matrix and the proposal for an assessment model using the LSTM network to create the assessment model of influencing factors of teaching quality in colleges and universities. Section 4 discusses the experimental results observed from the analysis of the proposed implementation. Section 5 presents the current work's conclusion and future work's development.

## 2. Related work

Wang and Zhang [14] examined the application of Fuzzy Mathematics and Machine Learning (FM2L) algorithms in teaching quality of college assessment models to create a more thorough, reasonable, and accurate assessment of classroom teaching in college education with the quality of university teachers. Focuses on developing and applying performance evaluation tools for educational information technology that utilize neural network simulation tools and MATLAB. The test results' margin of error is less than 5%, and the evaluation outcomes have a high baseline value. The findings demonstrate the accuracy and viability of the BP neural network performance evaluation method and serve as a critical benchmark for decision-making in educational management. Experts' experience, knowledge, and judgment are used to analyze the indicators of quality education for the second-level assessment evaluation index of information technology teachers, general teachers, and digital source libraries.

Rodríguez-Hernández et al. [15] aimed to test a systematic procedure for implementing artificial neural networks to predict academic performance in higher education. The sample included 162,030 students from Colombia's private and public universities. Results show that artificial neural networks can classify students' academic performance as high or low, with an accuracy of 82%. They outperform other machine-learning algorithms in evaluation metrics like recall and F1 score. Prior academic achievement, socioeconomic conditions, and high school characteristics are also important predictors. The study concludes with recommendations for implementing artificial neural networks and analysis considerations.

Costa-Mendes et al. [16] compared a multilinear regression model with machine learning algorithms using an anonymous 2014–15 school year dataset from the Portuguese Ministry of Education. The model is combined with random forest, support vector machine, artificial neural network, and extreme gradient boosting machine stacking ensemble. A hybrid analysis is designed to enhance the predictive ability of the machine learning algorithms. The article suggests that an information system supporting the nationwide education system should be designed to collect accurate data about academic achievement antecedents. No evidence is found in favor of smaller classes.

Feng and Feng [17] utilized the concept of an improved genetic algorithm; the initial limit and the weights of the Fuzzy BPNN are enhanced for identifying the multimodal quality of teaching factors by using an Adaptive Variation Genetic Algorithm with an optimized (AVGA-FBPNN) model created. A three-combined teaching quality assessment model based on the entropy calculation is used to avoid unnecessary subjective measures of teaching quality indices consideration. The findings demonstrate that the model can resolve the issues

associated with excessive subjectivity measures, overfitting, and slow convergence. It proves the model's efficacy in addressing the colleges and university's assessment of teaching quality accuracy is 0.97%. Utilized several evaluation stages like a task, student-teacher interaction, assignments, student attention, and lecture duration.

Tetteh and Agyei [18] employed the approach of proportional stratification of Sampling Randomization and Exploratory Data Analysis (EDA) to evaluate pre-service teachers' quality in mathematics education in colleges. The findings revealed that pre-service teachers believed that factors of learning-teaching environments, inspiration, parents' educational backgrounds, and the skilled expertise of mathematics affected their performance. It was also shown that teaching-learning environments and pre-service teachers' enthusiasm were the best indicators of how well they would achieve in mathematics. All the sampled colleges had a Kaiser-Meyer-Olkin (KMO) evaluation score of 0.849, suggesting that the sample size was enough for the factor analysis. Pre-service teachers thought their performance was influenced by the learning-teaching environment, motivation, parents' educational status, and mathematical skills.

Das et al. [19] introduced an automated system for generating multiple-choice test items with distractors. The system selects informative sentences using topic-words, chooses the best keyword as an answer key, and then transforms the sentence into a question-sentence. The system generates incorrect options or distractors using a feature-based clustering approach, without relying on external information or knowledge-base. This system demonstrates the efficiency of generating MCQs with distractors, making it a valuable tool for assessment in competitive examinations and contemporary information.

Chen et al. [20] suggested a DL-based Enhanced Learning Support Vector Machine based on (DL-ELSVM), a simulation model for evaluating university teaching effectiveness. Implementing a data mining algorithm generates an evaluation model, the procedure for the evaluation process, and optimization to increase the precision with which teaching quality in colleges and universities is evaluated. Encourages teachers to include virtual simulation experiments in the curriculum to improve how students learn. The outcomes demonstrate that this approach is a high accuracy and efficiency approach with 0.004% MSE, and it takes 0.249s of CPU time to evaluate teaching quality. Applied teaching quality evaluation based on teachers' content knowledge, instructional strategies, and pedagogical practices.

He et al. [21] applied the Back Propagation Neural Network and Fuzzy Mathematics (BPNN-FM) model to develop a system of assessments for intelligent teaching ability that helps college students assess their ability to research following knowledge acquisition. It is based on the BP neural network and fuzzy mathematics. A single deep hidden layer is utilized as the evaluation model; the BP neural network uses a level 2 index for the neuron count in the layer that receives input and an evaluation index for the neuron count in the layer that produces output. The results show that the minimum MSE has various convergence rates and better recognition network rates. Evaluation based on student grades like medium, good, and excellent, professionalism, research involvement, and problem-solving skills.

Mehta et al. [22] presented a three-dimensional DenseNet self-attention neural network (DenseAttNet) to identify and evaluate student participation in modern and traditional educational programs. The proposed DenseAttNet model outperformed all existing methods, achieving baseline accuracy of 63.59% for engagement classification and 54.27% for boredom classification. It also outperformed DenseAttNet trained on all four multi-labels, achieving accuracy of 81.17%, 94.85%, 90.96%, and 95.85%, respectively. The model achieved the lowest Mean Square Error (MSE) value of 0.0347 in a regression experiment on DAiSEE, and a competitive MSE of 0.0877 when validated on the Emotion Recognition in the Wild Engagement Prediction (EmotiW-EP) dataset.

Xi [23] introduced the Dragonfly Algorithm to the BP network (DA-BP) network to effectively evaluate the English academic instruction with audio-visual quality of language courses for speaking, listening, and reading. The evaluation indices consider teaching attitude, content, method effect, research ability, professionalism, and students' feedback as performance indicators. The input is taken from the 2018–2019 data, including 235 data details for research evaluation. The result proved that the proposed algorithm gives a minimum RMSE value of 0.07% and fast network convergence for training results. The major limitation is that the network performance depends on the number of nodes in the hidden layer. As performance indicators, the assessment indices consider teaching attitude, material, method effect, research skill, professionalism, and student feedback.

Yağcı [24] proposed a new machine learning model to predict final exam grades of undergraduate students using midterm exam grades. The model uses various algorithms such as random forests, nearest neighbour, support vector machines, logistic regression, Naïve Bayes, and k-nearest neighbor. The dataset consisted of 1854 students who took the Turkish Language-I course at a state university in Turkey from 2019–2020. The model achieved a classification accuracy of 70–75%, using only three parameters: midterm exam grades, Department data, and Faculty data. This data-driven study is crucial for establishing a learning analysis framework in higher education and contributing to decision-making processes. The study also contributes to early prediction of students at high risk of failure and determines the most effective machine learning methods.

Gong and Wang [25] suggested a paradigm for smart, data-driven assessment of the impact of teaching that depends on Fuzzy Comprehensive Analysis (FCA). Business data, such as teachers' performance, educational resources, and student feedback, are promptly gathered through courses taken online. Based on the technological design, simulation tests are carried out in which the proposal's technological architecture is implemented on a created Digital platform. The results revealed 94% and 94.3% correlation coefficients, respectively. The results demonstrated a strong degree of proximity and a significant relationship between the outcomes of the measurement tools, and the slopes of the linear regression model were very close to "1". Evaluated by taking online classes, information about teachers' performance, instructional materials, and feedback from students is quickly acquired.

The Google Scholar database is used to create the journal articles that will be considered for the literature study. The keywords "teaching quality" AND "college and university" are employed. Only journal papers published between 2019 and 2023 were collected using a filter for the year of publication. The selected elements of teaching quality were then evaluated in the journal publications to see if they apply to higher academic programs in colleges and university settings. Based on the survey, there are several issues in attaining high accuracy, entropy measure, and MSE. The various algorithms analyzed from the related work still have some limitations. It compares various metrics like accuracy, entropy measure, MSE, TLI, and other parameters for assessing influencing factors of teaching quality among colleges and universities. The algorithms AVGA-FBPNN, DL-ELSVM, CFNN, and DA-BP are taken for comparison purposes to showcase the validity of the proposed SIF-LSTM model [26].

## 3. Sequential Intuitionistic Fuzzy for assessing the membership degree of factors

Universities and colleges have gathered vast data for years to assess education and teaching quality. Colleges and universities should fully comprehend the significance of teaching quality, apply effective strategies to promote teaching quality and monitor and assess it to ensure that the instruction meets the student's progress needs. A DL model for gathering and analyzing

factors influencing the enhancement of college teachers' academic and technical capabilities was proposed to raise the innovation of academic achievement in colleges and universities and assist teachers in improving their knowledge related to academic growth. DL can also be used to identify the most effective teaching methods for specific subjects or topics. By analyzing large datasets of student performance data in different subject areas, deep Algorithms for learning can spot relationships and trends that human experts may not immediately recognize. Thus developing more effective curriculums and instructional strategies and improving educational outcomes. Students who undergo assessments can better reflect on their learning, and teachers who underwent a review may be able to improve their teaching techniques.

### 3.1 Assessment model based on intuitive fuzzy information theory

From Fig 1, finding the influencing factors pertinent to teaching quality is the first step in applying FIT. The assessment model includes teaching involvement, which refers to the period, effort, and passion teachers devote to their students' education, teaching, and professional growth. The phrase "combination" denotes that faculty members' "teaching involvement" entails integrating the above-discussed items related to teaching quality in colleges and universities.

The teacher's communication ability, technical aspect, size of the learning environment, the number of students, the teaching assets accessible to teachers and students, and the degree of student engagement are possible basic factors. These factors can be described as a fuzzy set that accurately conveys how much of an impact they have on the calibre of teaching.

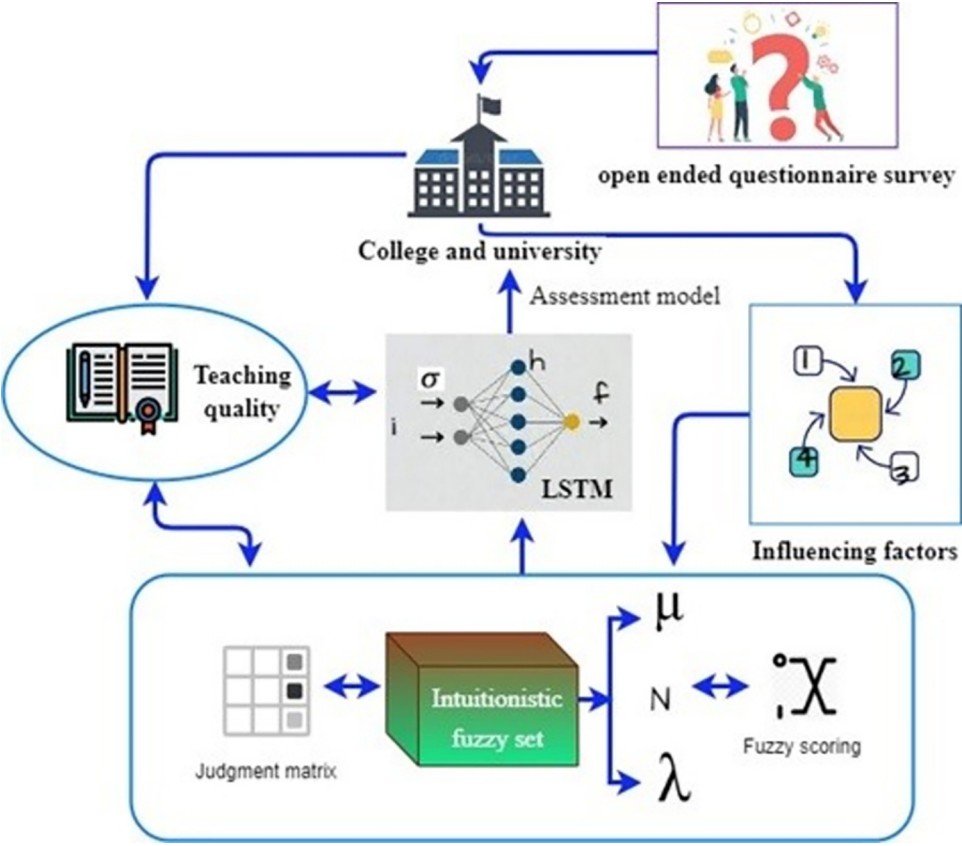

**Fig 1. Proposed scheme implementation.**

Step 1: Fuzzy set with intuition: Comprehending the fundamental concepts and techniques behind intuitionistic fuzzy sets is essential for comprehending fuzzy set theory for modelling the teaching quality assessment. The teaching quality outcome may be student performance, retention period, and feedback. Table 1 gives the details about the intuitionistic fuzzy set for influencing factors $I$ is applied to a college or university quality of teaching environment set $S$ is defined in Eqs (1) and (2) as follows:

$$I = \{i_1, i_2, \ldots, i_x\} \tag{1}$$

$$S = \{s_1, s_2, \ldots, s_y\} \tag{2}$$

Fig 2 describes the various influencing factors included in the questionnaire to assess the teaching quality among universities and colleges. The main categories used for evaluating teaching qualities are workload, skill competence of the teacher, course material readability, student-teacher interaction, classroom maintenance, teaching content, and materials. Each influencing factor is assessed with assessment qualities that must be considered for effective teaching quality evaluation. Initially, the teaching workload involves lesson preparation, student assessment, and class schedule. Teaching skill competence involves comprehensive quality and proficiency and a teaching attitude. Course material readability involves university rules and regulations and university curriculum design. Student-teacher interaction must consider social behaviour, support, and student motivation. Classroom maintenance centred around discipline and behaviour maintenance among the students inside the classroom. Teaching content and materials depend on the technology involved, like presentations and live video examples usage while teaching. The sample influencing factors are means of teaching: The factors in this sense need to take into account if the teaching methods satisfy the demands of contemporary, versatile teaching, the teaching methods are unique in terms of using AI, Virtual Reality (VR), laptops, interactive learning environment, recording devices, and other

**Table 1. Assessment of teaching quality of different levels of influencing factors in colleges and universities.**

| The intuitive fuzzy number for influencing factors ($n$) | First-level assessment qualities | Second-level assessment qualities |
|---|---|---|
| $i_1$ | Workload | Time spent, no. of courses, lesson preparation, answering questions, and student assessment |
| $i_2$ | Skill competence | Level of teaching, easy to understand, comprehensive quality, and proficiency |
| $i_3$ | Energy investment | Mental strength, amount of hard work |
| $i_4$ | Emotional involvement | Attitudes and feelings, moral support, passion, and parent-teacher communication. |
| $i_5$ | Classroom maintenance | Classroom behaviour, discipline, learning atmosphere, gaining knowledge, and class schedule. |
| $i_6$ | Course material readability | Knowledge acquisition, university curriculum matching, and professional exposure. |
| $i_7$ | Student-teacher interaction | Relationship, communication, doubt-clearing session |
| $i_8$ | Teaching methods | Q & A, assignments |
| $i_9$ | Teaching content | Proper content delivery, innovation, and motivational |
| $i_{10}$ | Mentoring | Coaching, instructing, collaborating, encouraging |
| $i_{11}$ | Flipped Classes | Prioritizing active learning and pre-recording the lectures. |
| $i_{12}$ | Assignments | Increasing performance level, engagement, and learning tasks |
| $i_{13}$ | Dedication | Curiosity of the learner and extracurricular activities |
| $i_{14}$ | Entrepreneurship | Self-confidence, risk-taking, creative thinking |

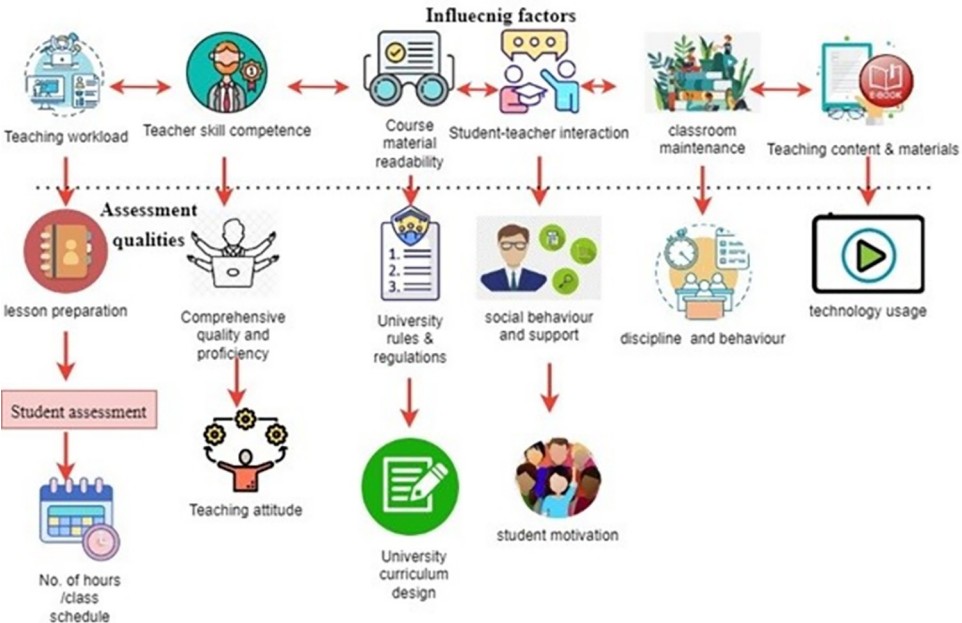

**Fig 2. Schematic diagram of influencing factors of teaching quality.**

technical equipment; that the teaching has feasibility, not only for demonstration; and the rate of implementation of smart teaching methods. According to the real teaching environment, teachers can suitably enhance the significance of academic achievement modules and ultimate test results. In Fig 2, the influencing factors of teaching quality are evaluated from the open questionnaire survey mentioned in the dataset. The survey responses are collected from the students and evaluate teachers' teaching quality in a university.

Student's feedback for improvement: Feedback is a type of knowledge a student gives regarding the various facets of the receiver side performance, which has a significant achievement impact. Feedback studies in education have since come to light, demonstrating the beneficial correlation between student feedback and quality of teaching effectiveness. Particularly, student feedback inspires more performance-focused actions to improve teachers' teaching abilities.

College and university culture impact: An institution's adoption of teaching- and learning-centred strategies favourably impacts the interaction between educators and learners, raising the standard of teaching in the learning environment. They highlighted that a way of life with better teaching quality would probably result in better learning and student engagement. An academic entity's entrenched structures, actions, common beliefs, principles, and ideology are called its college and university development culture.

Step 2: Create intuitionistic fuzzy information sets to depict the strength of the relationships between every factor that impacts each outcome of teaching quality in college and university. To describe the ambiguity and insufficient information in the college details, use the three variables in intuitionistic fuzzy information sets: degree of membership ($\mu$), degree of non-membership ($N$), and degree of hesitation ($\lambda$). Following the assignment of membership values, the overall effect of each factor on TQ can be assessed using a set of measurable metrics. These metrics could consist of variables like assessments of teacher quality, parent and academic administration critiques, and student accomplishment scores.

Step 3: Apply fuzzy inference rules like AND, OR, and NOT to assess the level of correlation between each influencing element and all the outcomes related to teaching quality. Eq (3)

defines the scoring function $f(n)$ may be utilized for assessing an intuitive fuzzy number $n$.

$$f(n) = (\mu_I(n), s_y(n)) \tag{3}$$

The matrix for the teaching quality assessment is derived in Eq (4) as follows:

$$\mu(i_1 \rightarrow s_1) = [\mu(i_1) * \mu(s_1)] + \lambda(i_1, s_1) * [1 - N(i_1) * N(s_1)] \tag{4}$$

Where $N(i_1)$ denotes the non-membership function that negates the influencing factor from the membership function. $N(i_1) = 1 - \mu(i_1)$. The $\mu(i_1)$ denotes the extent to which influencing factor $i_1$ is the fuzzy set member $I$. $\mu(s_1)$ denotes the extent to which teaching quality $s_1$ is the fuzzy set member $S$. $N(i_1)$ is the extent to which the influencing factor $i_1$ is not a fuzzy member set $I$. $N(s_1)$ is the extent to which teaching quality $s_1$ is not a fuzzy set member $S$.

Step 4: Construct a judgment matrix ($J$) based on intuitive fuzzy information theory based on evaluation indices is calculated in Eq (5) as follows:

$$J = (j_{12}{}^x)_{n \times n}, \text{ where } j_{12}{}^x = (\mu_{12}, N_{12}) \tag{5}$$

Where $j_{12}{}^x$ represents the judgment matrix for an individual influencing factor of the first and second element that is $i_1$ and $i_2$. The decision-maker of influencing factors for particular teaching quality typically arrives at an intuitive fuzzy figure inferred from experience and real-world issues. After determining the weights for each influencing indicator factor, conduct a thorough analysis of the assessment's findings using the observed data analysis. To calculate the affiliation degree for each level separately, determine the assessment judgment matrix. By contrasting each assessment index's real value with its standard value, it is possible to determine the affiliation degree. Then, to fuzzily the data to smoothly transition between the levels and prevent data jumps of correlations among the influencing factors. Ultimately, the scoring levels are determined by deriving the final ratings on students' performance feedback.

## 3.2. Prediction of assessment model using LSTM

As discussed in the previous section, the fuzzification of influencing factors that assess the teaching quality impact is recognized. The degree is correlated using the LSTM model, and its detailed discussion is given in the current section, step by step, in Fig 3.

Step 5: Construct a DL model, such as an LSTM, using the intuitionistic fuzzy information sets as inputs to predict and classify the results of the quality of teaching among colleges and universities based on the influencing components. By entering this fuzzy-based information as input into the LSTM model, it might determine what factors are most closely linked to teaching quality achievement and, in turn, identify the main forces influencing teaching quality in colleges and universities. As a more effective Recurrent Neural Network (RNN) model, LSTM can better handle sequences of various lengths of influencing factors. It is common practice to employ LSTM for the assessment of teaching quality.

$$in_g = \sigma(wt[i_t, h_{t-1}] + b_i) \tag{6}$$

$$fr_g = \sigma(wt[fr_t, h_{t-1}] + b_{fr}) \tag{7}$$

$$o_g = \sigma(wt[o_t, h_{t-1}] + b_o) \tag{8}$$

$$f_g = tanh(wt[f_t, h_{t-1}] + b_f) \tag{9}$$

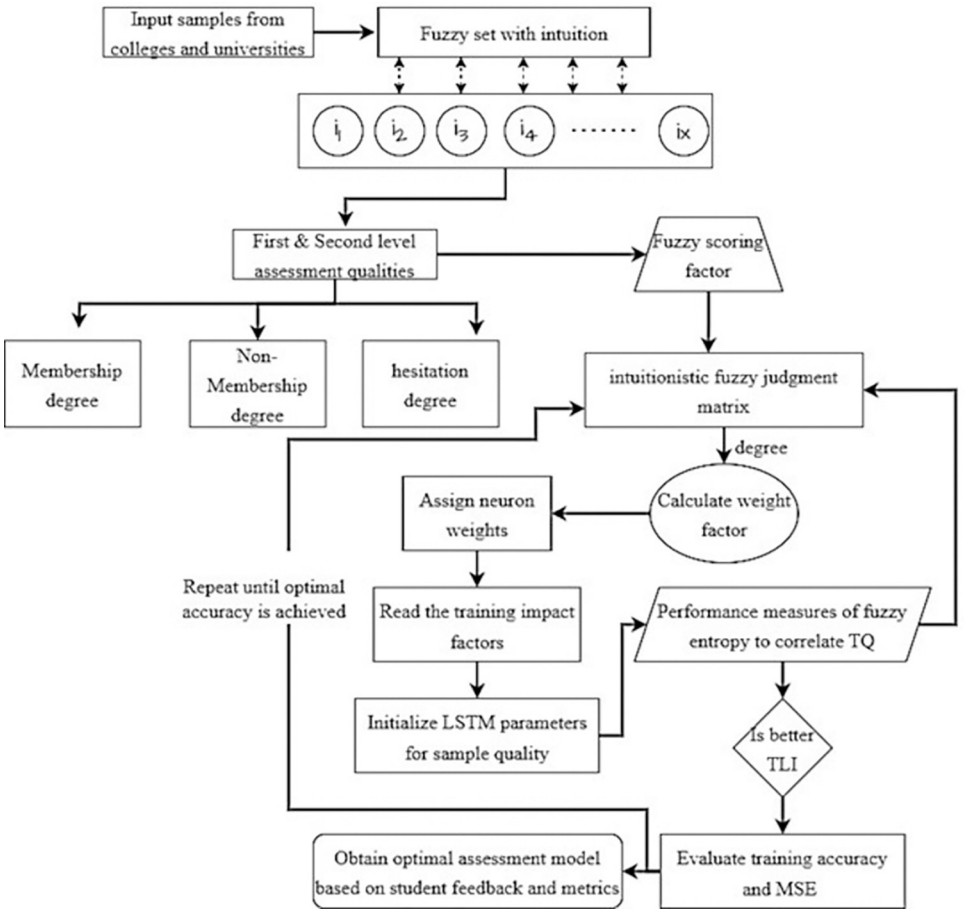

**Fig 3. Proposed SIF based on LSTM model for predicting TQ assessment.**

$$c_g = fr_g \odot c_{g-1} + in_g \odot o_g \qquad (10)$$

$$h_g = \tanh(c_g \odot o_g), h = \begin{pmatrix} i_x \\ h_{t-1} \end{pmatrix} \qquad (11)$$

Where $in_g, fr_g, o_g, m_g, c_g$ and $h_g$ the input, forgotten, output, final cell modulation, input memory cell modulation, and hidden state of the LSTM cell with corresponding bias are given as $b$ are calculated from Eqs (6)–(11). The activation function sigmoid is represented by σ. Similarly, wt represents the weight matrices with corresponding cells for all the influencing factors. $h_{t-1}$ denotes the previous hidden state of the LSTM cell, which directly represents the influencing factor that is highly correlated with teaching quality.

The training impact of the LSTM network, which is directly influenced by parameter setting, is the sole factor in this educational environment that influences the prediction outcomes of different teaching quality measures.

Step 6: By contrast the predictions of the DL model with the actual results of teaching quality, the assessment of the teaching quality model's performance. To assess the level of uncertainty, reluctance, or hesitation in the predicted model, employing intuitionistic fuzzy

measures such as the intuitionistic fuzzy entropy can be calculated using necessary formulas in the results and discussion section. The intuitionistic fuzzy divergence is a metric that quantifies the difference between two fuzzy sets with intuition. In contrast, the intuitionistic fuzzy entropy formula determines the degree of unpredictability in a piece of intuitionistic fuzzy information set. Intuitionistic fuzzy values, categorized by membership and non-membership and hesitation degrees, are the basic elements of intuitionistic fuzzy sets (IFS). If each membership of a fuzzy set defined in a domain of discourse is a 4-tuple consisting of membership degree, hesitation degree, and non-membership degree, then the set is termed an intuitionistic fuzzy set (IFS). It is uncertain if the degree of hesitancy degree toward membership or non-membership.

Step 7: Performance Assessment: Based on the assessment findings and student feedback related to a questionnaire, enhance the model by changing the parameters of the DL model and fuzzy sets with intuition. Repeat steps 3 to 6 until the accurate fuzzy model is obtained.

The proposed model SIF-LSTM makes it possible to implement corrective measures to develop better teaching approaches or pedagogical structures that encourage more effective learning. These results provide crucial information for colleges and universities improvement decisions, advantageous plans for creating an environment that values excellence in teaching within the institution, setting up student composition plans, and resource allocation decision-making for fostering academic resources to encourage teaching quality. Teachers can better grasp the complicated relationships essential to effective teaching using intuitionistic fuzzy sets to assess teaching quality-influencing factors.

## 4. Experimental result analysis and discussion

The experimental analysis is organized by arranging the data sources' description and performance of assessment matrices by comparing graph analysis with existing literature algorithms. The technical roadmap of the paper includes the rapid teaching quality assessment, quality assurance, visualization of the student outcome and technical knowledge.

### 4.1. Detailed description of data source

Collecting enormous amounts of data can improve the spatial precision of calculations. Analyzing college students' learning effects from a teaching perspective requires accurate and trustworthy data. The methods utilized to gather the data for this study and the type and format of data need to be considered. Generally speaking, two different forms of information are used for analysis: precise and fuzzy input. Fuzzy-related techniques typically include scores, times charts, mathematical written works, designs, and other forms. However, one may typically derive the outcomes for educational impacts for students in higher education from the data in visualizations, designs, writings, etc. Precision data refers to information capable of describing the TQ for college students. In contrast, the term "fuzzy data" describes values for parameters that are challenging to ascertain. The set of influencing factors of teaching quality in undergraduate education in China is analyzed and validated with an open-ended questionnaire survey from the teacher and student perspective, which is considered a data source in this study. Here, the main categories that need to be considered for teaching qualities are workload, ability, energy, and teaching emotional investment. In addition, the other parameters that impact teaching quality that need to be considered are classroom management, clarity of teaching course materials, student-teacher interaction, teaching strategies (methods, content), skills, and work engagement. The second-level assessment qualities related to this teaching quality are discussed in Table 1.

**Table 2. Findings related to influencing factors of teaching effects in colleges and universities.**

| Influencing factors | A major dimension of teaching qualities |
|---|---|
| Workload | Time spent, no. of courses, lesson preparation, answering questions, and student assessment |
| Skill competence | Level of teaching, comprehensive quality |
| Energy investment | Mental strength, amount of hard work |
| Emotional involvement | Attitudes and feelings, passion |

An open dataset of 65 university students encompasses an institute's social sciences, the humanities, and science and engineering, and 62 faculty members with various university levels and types, professional positions, teaching demographics, and designations participated in a questionnaire with an open-ended poll [26]. The initial influential factors of workload involvement, competence, energy investment, and teaching emotional involvement make up most of faculty members' teaching contributions. The suggested four-dimensional teaching involvement model was done for a preliminary test on 342 faculty members. Confirmatory component analysis revealed that the data assessment model fit the test's 293 faculty participants well. A college education is crucial for academic achievement, and out of the elements influencing its quality, the most significant factor is the quality of the teaching. Currently, a significant problem impacting teaching quality is a limited educational investment. Faculty members' four-dimensional teaching involvement is strongly associated with handling the classroom, the readability of the course materials, teacher-student contact, the use of teaching techniques and skills, and work satisfaction are discussed in Table 2. The level of teaching involvement made by college teachers teaching quality in colleges and universities increased as the score increased.

The proposed SIF-LSTM model can predict the quality of teaching in colleges and universities and produce improved results for any new set of influencing factors. The open-ended questionnaire survey responded to by student feedback from has been considered for this. The student feedback on teaching quality questionnaire survey results is utilized in sequential intuitionistic fuzzy entropy for analyzing the correlation between various influencing factors given by them. It is taken as input to a fuzzy set parameter that can be adjusted intuitively to improve the DL model. The intuitionistic fuzzy entropy indicates the model's reliability by establishing a correlation between the elements collected from the dataset that significantly influence teaching quality using membership values and fuzzy scores. Three states are analyzed to obtain the greatest degree of intuitionistic fuzzy entropy value. The weighted influencing factors of teaching quality are given to the LSTM model for training and to categorize the factors that affect the TQ using the TLI measure. With the help of a trained LSTM model, the factors affecting the teaching quality can be accurately evaluated.

## 4.2 Comparative performance analysis of experimental results

As mentioned in the algorithm steps, assessing influencing factors of college and university teaching effects using proposed techniques to calculate the intuitionistic fuzzy entropy is a valuable metric for assessing the accuracy and precision of the model's predictions. The proposed assessment model can forecast college and university teaching quality outcomes based on any number of new groups of influencing factors once it has been trained. The level of unpredictability in these forecasts can be assessed using the intuitionistic fuzzy entropy (*en*) from Eq (12), which provides an index of the model's reliability. Actual teaching quality outcomes versus the projected outcomes from the model to create a comparison graph relating to

predicted outcomes for the quality of teaching. For an instance of intuitionistic fuzzy set, A, the intuitionistic fuzzy entropy has been defined in Eq (12) as follows:

$$en(I, S) = -[\mu(I, S)*\log_2(\mu(I, S)] + N(I, S)*\log_2(N(I, S))] \tag{12}$$

The membership degree ranges from 0 to 1, where 1 denotes complete set membership, and 0 denotes no set membership for the influencing factors that affect the teaching qualities in a learning environment. A set of projected teaching quality outcomes will have a high intuitionistic fuzzy entropy value if there is considerable ambiguity or unpredictability in forecasting the model. It shows that the assessment model has to be enhanced or that further data is required to lower uncertainties.

Fig 4 shows the intuitionistic fuzzy entropy of the proposed assessment model for identifying the degree of relationship among influencing factors impact on teaching quality rises as the hesitation degree does with the $\mu(I)$ and $N(I)$. Likewise, in the non-membership function, 0 denotes the influencing factor that fully affects the teaching outcome set, and 1 represents that it does not affect any teaching quality factors. The degree of membership is an essential and sufficient criterion for reaching the highest value of intuitionistic fuzzy entropy. The lack of certainty caused by the degree of hesitation with a non-membership degree is greatest when the degree of non-membership is 0 or 0.5 and when the degrees of membership and non-membership are both 0. The biggest degree of ambiguity brought on by the fuzziness occurs when the degree of membership and non-membership is 0.5.

The Tucker-Lewis Index (TLI) is an evaluation metric that can be utilized to examine the factors influencing college and university teaching quality as defined in Eq (13). The chi-square scores and dimensions of independence of the hypothesized paradigm and a null model with no correlations between the factors in question might then be compared using the

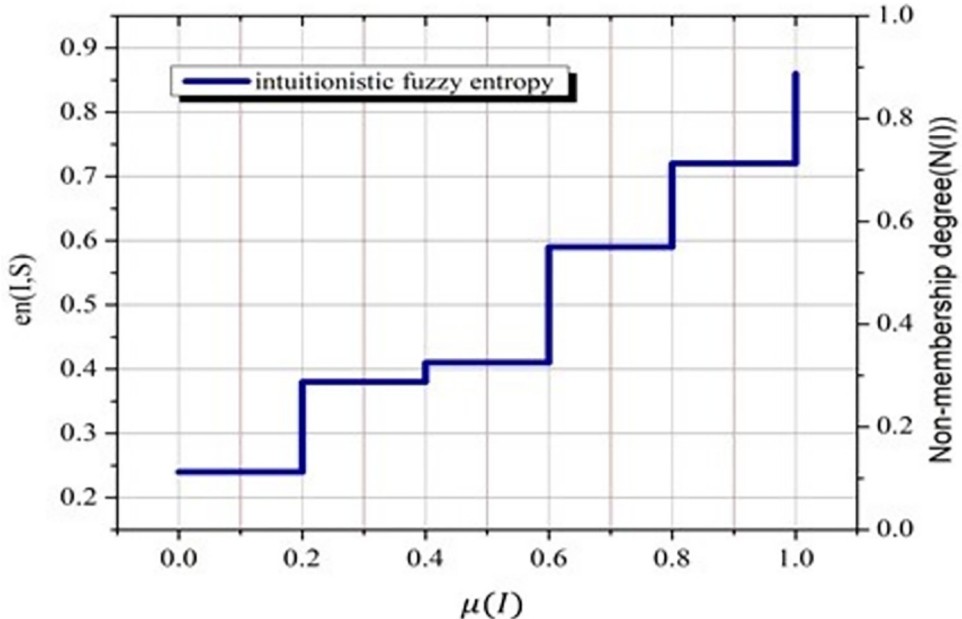

**Fig 4. Implementation of IF for *en(I,S)*.**

TLI to assess the model's fit.

$$TLI\ test = \frac{(\alpha^2 null - \alpha^2 model + Cnull - Cmodel)}{(\alpha^2 null + Cnull)} \tag{13}$$

Where $\alpha^2 null$ denotes the chi-square value of influencing factors with no relationship between them. $\alpha^2 model$ denotes the model to be tested. Cnull represents the null model degree of fitness. Cmodel denotes the hypothesis testing model.

Fig 5 discusses the influential factors that impact the measurement of teaching quality using TLI measure with the help of Eq 5. The TLI ranges from 0 to 1, with values closer to 1 suggesting that the model fits the data more accurately, and less than 0.5 indicates the partial correlation of the model fits the data. Using the TLI analysis results to determine the factors influencing college teaching quality most. To more accurately represent the fundamental factors that affect teaching quality, teachers could utilize the assessment to modify the structure of the relations by including or eliminating factors. By implementing the proposed SIF-LSTM model, the TLI measure analyzes the influencing factors very well before spoiling the entire learning environment of academic institutions with a range of 0.81% compared to other existing approaches. Then the LSTM model parameter setup is defined in Table 3.

The effectiveness of the suggested strategy is assessed for solution quality using a formula that measures calculation accuracy. True Positive (TP) denotes the factors correctly correlated with the influencing element, and True Negative (TN) denotes the wrong assessment prediction in teaching quality. False Positive (FP) denotes the imperfect prediction of a factor that does not influence the teaching behaviours of a particular faculty in the college. False Negative (FN) denotes the correct mismatch prediction of the non-membership degree of influencing factors. The calculation of the accuracy of the assessment model is described in Eq (14) as

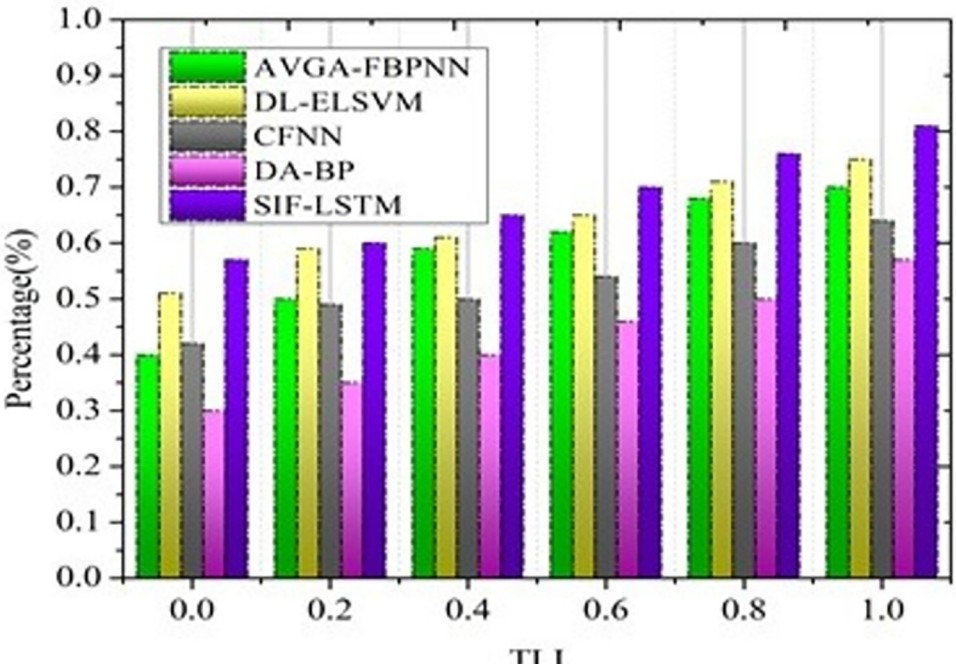

**Fig 5. TLI measure for influential factors.**

**Table 3. Parameter setup of the LSTM model.**

| Sl. No | Parameter Names | Value |
|---|---|---|
| 1 | No. of hidden layers | 100 |
| 2 | Loss function | MSE |
| 3 | The learning rate for influencing factors | 0.005 |
| 4 | Maximum threshold limit | 1 |
| 5 | Intuitionistic fuzzy number | 9 |
| 6 | Maximum rounds for training | 150 |
| 7 | Epochs | 50 |

follows:

$$Accuracy = TP + TN/TP + TN + FP + FN \qquad (14)$$

From Fig 6, the training accuracy results acquired from the proposed model give a better 98.4% compared to other models AVGA-FBPNN, DL-ELSVM, CFNN, and DA-BP. The training accuracy is checked for optimal input influencing factors. If the result is not achieved, then the input parameters need to be fine-tuned by adjusting the LSTM model initial settings of all input to final states with the necessary sigmoid function. Repeat the steps from the judgment matrix to update and enhance the accuracy of the predicted assessment model.

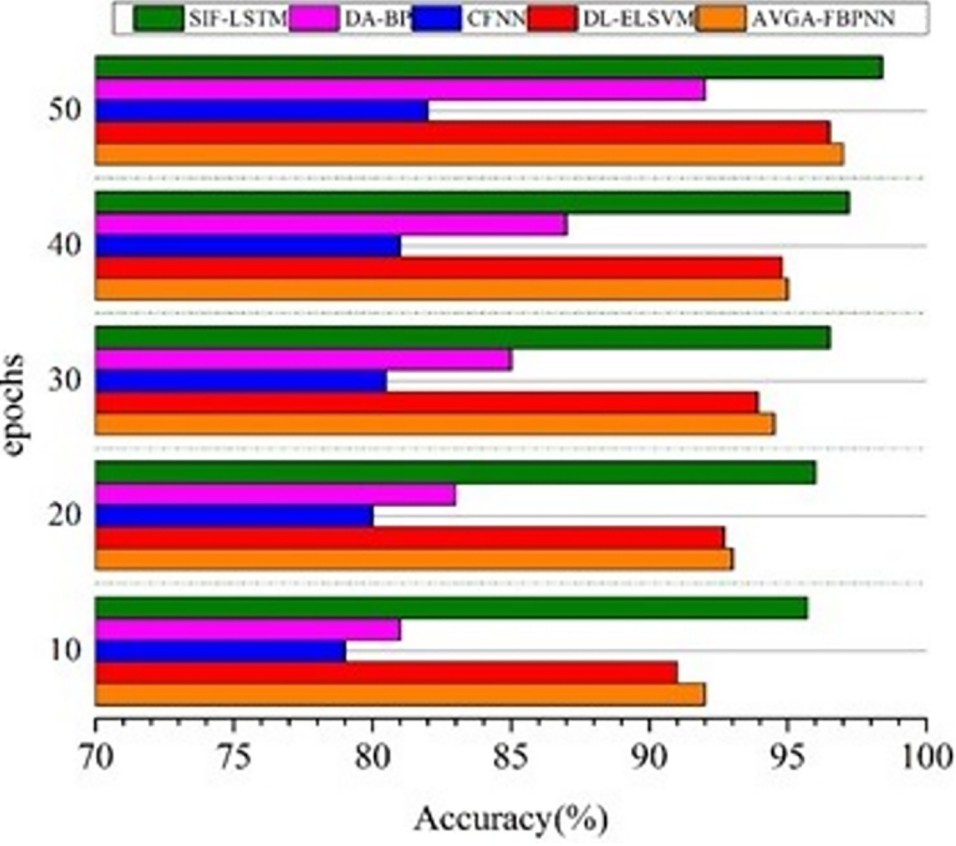

**Fig 6. Accuracy comparison of various algorithms.**

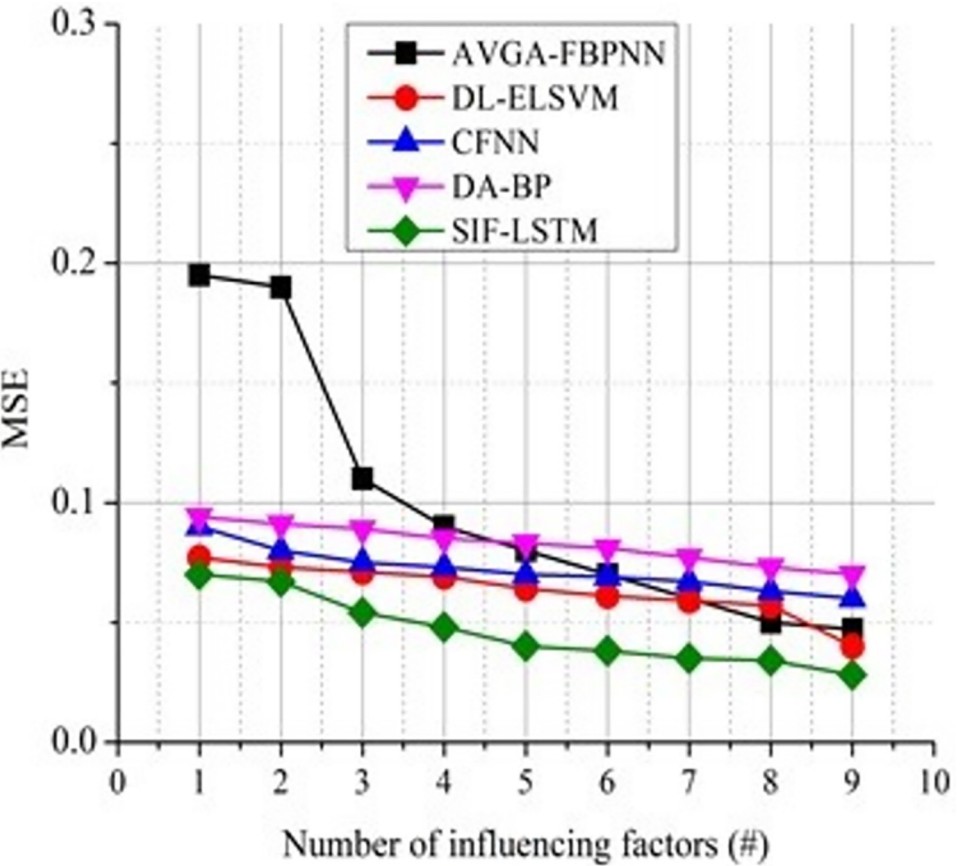

**Fig 7. Mean square error for various influencing factors.**

The mean squared variance among the expected and actual teaching quality scores defines the statistical measures. Table 3 describes the loss function of the training and validation model for the evaluated quality factors that can be identified in Eq (15).

$$MSE = \left(\frac{1}{n}\right) * sum(expected - \hat{actual})^2 \tag{15}$$

Actual value of teaching quality of $n^{th}$ teacher working in the university is taken as a sample, and it is compared with the predicted value of teaching quality with the corresponding influence factor, $n$ represents the total number of teachers quality considered.

From Fig 7, the MSE value gives various results for influencing factors in which the proposed algorithm surpasses the remaining algorithms based on the mathematical formula in Eq (15). The teaching quality is influenced by various factors, such as teaching modes and methods, classroom maintenance, workload, skill competence, etc., as mentioned in Table 1. The MSE of the proposed algorithm gives 0.028, and the remaining algorithms, like AVGA-FBPNN, DL-ELSVM, CFNN, and DA-BP, give somewhat greater error prediction values. Table 4 shows the effectiveness of the proposed SIF-LSTM.

The research results are summarized as follows: The experimental result analysis shows that the proposed algorithm performs better than other existing algorithms in terms of MSE, accuracy, entropy measure, and TLI index values. The proposed algorithm SIF-LSTM gives the best assessment model for various influencing factors of teaching qualities of faculties in

**Table 4. Effectiveness of the proposed SIF-LSTM.**

| Parameter | AVGA-FBPNN | DL-ELSVM | CFNN | DA-BP | Proposed SIF-LSTM |
|---|---|---|---|---|---|
| TLI measure | 0.07 | 0.08 | 0.06 | 0.05 | 0.028% |
| Accuracy | 96.3% | 95.6% | 82.3% | 91.4% | 98.4% |
| Mean Square Error | 0.04 | 0.05 | 0.08 | 0.09 | 0.03 |

**Table 5. Qualitative analysis of the proposed SIF-LSTM.**

| Parameter | AVGA-FBPNN | DL-ELSVM | CFNN | DA-BP | Proposed SIF-LSTM |
|---|---|---|---|---|---|
| Time investment for preparation | 78 hrs | 72hrs | 67hr | 54hr | 36hr |
| Time efficiency for assessment | 64hr | 60hr | 58hr | 50hr | 35hr |

colleges and universities. Based on the entropy measure of intuitionistic fuzzy value, it identifies a correlation between the factors that greatly impact the quality of teaching. Hence, it is recommended that the proposed model works well to improve academic institutions with better teaching qualities. In addition, SIF-LSTM system efficiency is further evaluated using qualitative metric such as time investment for preparation and time efficiency for assessment. The obtained result is shown in Table 5 and the results are compared with existing methods such as AVGA-FBPNN, DL-ELSVM, CFNN, and DA-BP.

From the Table 5, it clearly states the proposed SIF-LSTM approach attains minimum preparation and assessment time for improving the teaching quality. The set of training samples and patterns are more useful to understand the concept that leads to minimize the time consumption. However, the system consumes high teaching quality and efficiency compared to other methods.

## 5. Conclusion

The conclusion about the proposed work SIF-LSTM gives a few guidelines, implications, and potential directions for academic growth by analyzing the various factors affecting teaching quality. The various considerations of the teaching assessment calibre are taken in many universities and colleges, and their influencing factors are identified using intuitionistic fuzzy and DL-based LSTM networks. Experts then use intuitionistic fuzzy sets for assigning a value of membership to each factor. According to these membership values, every factor is thought to affect teaching quality to varying degrees, along with the unpredictability surrounding that notion. The major research limitation of the sequential combination of intuitionistic fuzzy information theory is that it does not address uncertainty, a judgment instability in expert judgments. It is suggested that in the future, colleges create apps that mix IoT and AI, intending to comprehend and forecast various factors that influence teaching effectiveness. Automating a speedy response from fuzzy neural network algorithms can allow better management of universities. It makes it possible to implement corrective measures to develop better teaching approaches or curricular networks that encourage more effective learning. The enhancement of the SIF-LSTM model for assessing the influencing factors of teaching quality is proved by the accuracy of 98.4%, Mean Square Error (MSE) of 0.028%, Tucker Lewis Index (TLI) measure for all influencing factors and entropy measure of non-membership and membership degree correlation of factors related to quality in teaching by various dimensional measures. In conclusion, this research suggests that future studies include teaching staff and students to provide a more complete picture of how the learning strategy has evolved, particularly concerning another crucial aspect of education.

## Author Contributions

**Investigation:** Jie Yu.

**Methodology:** Jie Yu.

**Resources:** Jie Yu.

**Visualization:** Jie Yu.

**Writing – original draft:** Jie Yu.

**Writing – review & editing:** Jie Yu.

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
