## [Decision Letter · Decision Letter 0]

12 Jan 2024

PONE-D-23-38898Evaluation of Influencing Factors of University Teaching Quality Based on Fuzzy Logic and Deep Learning TechnologyPLOS ONE

Dear Dr. Yu,

Thank you for submitting your manuscript to PLOS ONE. After careful consideration, we feel that it has merit but does not fully meet PLOS ONE’s publication criteria as it currently stands. Therefore, we invite you to submit a revised version of the manuscript that addresses the points raised during the review process.

We look forward to receiving your revised manuscript.

Kind regards,

Le Hoang Son, Ph.D

Academic Editor

PLOS ONE

Journal Requirements:

3. In the online submission form, you indicated that The datasets used and/or analyzed during the current study are available from the corresponding author on reasonable request.

Reviewers' comments:

Reviewer's Responses to Questions

**Comments to the Author**

1. Is the manuscript technically sound, and do the data support the conclusions?

Reviewer #1: Yes

Reviewer #2: Yes

Reviewer #3: Yes

2. Has the statistical analysis been performed appropriately and rigorously? 

Reviewer #1: Yes

Reviewer #2: Yes

Reviewer #3: Yes

3. Have the authors made all data underlying the findings in their manuscript fully available?

Reviewer #1: Yes

Reviewer #2: No

Reviewer #3: Yes

4. Is the manuscript presented in an intelligible fashion and written in standard English?

Reviewer #1: Yes

Reviewer #2: Yes

Reviewer #3: Yes

5. Review Comments to the Author

Reviewer #1: The abstract should be revised as it does not introduce the area of research and the research question.

The introduction should be rewritten professionally.

Please explain the proposed method in more detail, what is the novelty of the proposed method compared to the state of the art?

Current experiments are weak and a fair comparison with other recent methods is very necessary.

I think it would be good to represent the experiment results in the abstract.

The experiment description section is too rough.

A description of data collection should be added.

For the experimental results, it will be good to present a statistical test the comparison the results with other published methods.

This can help support the claim of improved results obtained with the studied selection methods.

Another aspect where the paper can be improved is motivation and the reason for the given architecture.

The current approach seems to be more like we have these different types of architectures, let's mix them and present results by training them.

It would be of great interest to why a particular model was selected and what part of the framework is helping us learn.

English can be improved. Proofreading should ensure the appropriate use of grammar, tense, and punctuation.

Longer sentences should be converted into smaller ones.

Reviewer #2: This study assesses the influencing factors of college teaching quality in China. The manuscript is well written and analyses seem robust. My major concern is the data collection, more details are needed regarding how the survey responses are collected, such as from where, whom, information about students and teachers’ background (major, what courses etc).

Specific comments:

I highly recommend the author including line number and page number for future manuscripts, this will allow reviewers to refer to comments more easily.

Title: I recommend including ‘China’ in the title if that is where the data are collected.

Abstract: should include more information on data, such as sample number, major results and implications.

Abstract: ‘approaches must be used …’ – change ‘must’ to ‘could’

Keywords: should use words that are not in the title

Introduction:

- ‘by numerous variables’: add references and explain why these variables are important

- ‘requirements can be encouraged’: replace ‘can be’ by ‘are’

- ‘Recognizes the components xxx’: this doesn’t seem to be a complete sentence

- ‘NN approaches’: specify what it is

- ‘FRs can effectively assess xxx’: add references

- ‘An influence factors analysis model xxx’: explain the performance of such model

- ‘teaching wushu’: what is wushu?

- Add references for ‘Resources for teaching wushu xxx’

Section 2 Related work has summarized too much literature, but not explaining why they are relevant to this work. If some don’t use the same method, I suggest removing them.

Figure 6: figure legend overlaps with some bars

Reviewer #3: This paper develops a powerful approach for assessing the influencing factors of college and university teaching effects by implementing the Sequential Intuitionistic Fuzzy (SIF) assisted Long Short-Term Memory (LSTM) model. The enhancement of the SIF-LSTM model for assessing the influencing factors of teaching quality is proved by the accuracy of 98.4%, Mean Square Error (MSE) of 0.028%, Tucker Lewis Index (TLI) measure for all influencing factors and entropy measure of non-membership and membership degree correlation of factors related to quality in teaching by various dimensional measures. The effectiveness of the proposed model is validated by implementing data sources with a set of 60+ teachers' and students' open-ended questionnaire surveys from a university.

However, there are some comments.

1- You must add qualitative comparison between your proposed scheme and some of existing methods based some metrices such as complexity time, methodology, preprocessing, and others.

2- In figure 3, the three boxes of membership degree, non-membership degree and hesitation degree do not connect any other box in the flowchart, I think there is something wrong.

3- the resolution of some figures needs some enhancements

6. PLOS authors have the option to publish the peer review history of their article (what does this mean?). If published, this will include your full peer review and any attached files.

Reviewer #1: **Yes: **Dr B Santhosh Kumar

Reviewer #2: No

Reviewer #3: **Yes: **Ahmed A. A. Gad-Elrab

---

## [Author Response · Author response to Decision Letter 0]

18 Feb 2024

Reviewer #1:

1. The abstract should be revised as it does not introduce the area of research and the research question.

Ans:

Abstract is rewritten accordingly.

2. The introduction should be rewritten professionally.

Ans:

Introduction section rewritten accordingly.

3. Please explain the proposed method in more detail, what is the novelty of the proposed method compared to the state of the art?

Ans:

The main intention of this study is to improve the teaching quality using SIF-LSTM method. 

A DL model for gathering and analyzing factors influencing the enhancement of college teachers' academic and technical capabilities was proposed to raise the innovation of academic achievement in colleges and universities and assist teachers in improving their knowledge related to academic growth. DL can also be used to identify the most effective teaching methods for specific subjects or topics. By analyzing large datasets of student performance data in different subject areas, deep Algorithms for learning can spot relationships and trends that human experts may not immediately recognize.

The excellence of the system is compared with existing methods such as AVGA-FBPNN, DL-ELSVM, CFNN, and DA-BP.

4. Current experiments are weak and a fair comparison with other recent methods is very necessary.

Ans:

The system is compared with existing methods AVGA-FBPNN, DL-ELSVM, CFNN, and DA-BP. During the comparison, qualitative and quantitative metrics are utilized to make the comparison.

5. I think it would be good to represent the experiment results in the abstract.

Ans:

Thanks for the positive response

6. The experiment description section is too rough.

Ans:

Experimental discussions are fine-tuned accordingly.

7. A description of data collection should be added.

Ans:

An open dataset of 65 university students encompasses an institute's social sciences, the humanities, and science and engineering, and 62 faculty members with various university levels and types, professional positions, teaching demographics, and designations participated in a questionnaire with an open-ended poll [26]. The initial influential factors of workload involvement, competence, energy investment, and teaching emotional involvement make up most of faculty members' teaching contributions. The suggested four-dimensional teaching involvement model was done for a preliminary test on 342 faculty members. Confirmatory component analysis revealed that the data assessment model fit the test's 293 faculty participants well.

8. For the experimental results, it will be good to present a statistical test the comparison the results with other published methods.

Ans:

statistical comparison is given according to your comment. Table 4 and 5 is utilized for this comparison.

9. This can help support the claim of improved results obtained with the studied selection methods.

Ans:

The results are improved accordingly and existing methods are utilized for comparison AVGA-FBPNN, DL-ELSVM, CFNN, and DA-BP.

10. Another aspect where the paper can be improved is motivation and the reason for the given architecture.

Ans:

The teacher's communication ability, technical aspect, size of the learning environment, the number of students, the teaching assets accessible to teachers and students, and the degree of student engagement are possible basic factors. These factors can be described as a fuzzy set that accurately conveys how much of an impact they have on the calibre of teaching. The main intension of this study is to improve the teaching quality according to various factors.

11. The current approach seems to be more like we have these different types of architectures, let's mix them and present results by training them.

Ans:

Here, LSTM utilized to train the data. According to the influencing factors information is trained that helps to improve the teaching quality. Therefore, present study is compared with the existing approaches to justify the proposed system efficiency. Here, existing system concept is taken how the method working on present study is analyzed to examining the system performance.

it's important to carefully design the ensemble, select appropriate architectures, and consider the integration of fuzzy logic and deep learning in a meaningful way. Additionally, the evaluation and interpretation of results should be thorough, considering the strengths and weaknesses of each architecture in contributing to the overall assessment of teaching quality.

12. It would be of great interest to why a particular model was selected and what part of the framework is helping us learn.

Ans:

the selection of the model and the components of the framework work synergistically to ensure that the analysis is tailored to the unique challenges and goals of evaluating university teaching quality based on fuzzy logic and deep learning technology. Each step in the framework contributes to a deeper understanding of the factors influencing teaching quality, ultimately leading to informed decision-making and improvements in educational practices.

13. English can be improved. Proofreading should ensure the appropriate use of grammar, tense, and punctuation.

Ans:

Document proofreaded.

14. Longer sentences should be converted into smaller ones.

Ans:

Checked longer sentences are changed into smaller ones.

Reviewer #2: This study assesses the influencing factors of college teaching quality in China. The manuscript is well written and analyses seem robust. My major concern is the data collection, more details are needed regarding how the survey responses are collected, such as from where, whom, information about students and teachers’ background (major, what courses etc.).

Specific comments:

I highly recommend the author including line number and page number for future manuscripts, this will allow reviewers to refer to comments more easily.

1. Title: I recommend including ‘China’ in the title if that is where the data are collected.

Ans:

Evaluation of Influencing Factors of China University Teaching Quality Based on Fuzzy Logic and Deep Learning Technology

2. Abstract: should include more information on data, such as sample number, major results and implications.

Ans:

The enhancement of the SIF-LSTM model for assessing the influencing factors of teaching quality is proved by the accuracy of 98.4%, Mean Square Error (MSE) of 0.028%, Tucker Lewis Index (TLI) measure for all influencing factors and entropy measure of non-membership and membership degree correlation of factors related to quality in teaching by various dimensional measures.

3. Abstract: ‘approaches must be used …’ – change ‘must’ to ‘could’

Ans:

Fuzzy and Deep Learning (DL) approaches must be could to build an assessment model that can precisely measure the teaching qualities to enhance accuracy

4. Keywords: should use words that are not in the title

Ans:

Keywords: Teaching quality, sequential intuitionistic fuzzy, predictive model, Intuitionistic fuzzy long short-term memory; membership degree; Influencing factors;

5. Introduction:

- ‘by numerous variables’: add references and explain why these variables are important

Ans:

Assessment of teaching quality is a challenging, unpredictable system issue influenced by numerous variables such as class size, learning environment, teacher-student relationship, learning outcomes etc.

- ‘requirements can be encouraged’: replace ‘can be’ by ‘are’

Ans:

Universities with various fundamental curriculum requirements are encouraged to use the binary relative assessment approach to analyze the influencing factors that affect the efficacy of teaching quality in experiment design across every university.

- ‘Recognizes the components xxx’: this doesn’t seem to be a complete sentence

Ans:

Recognizes the factors that influence the success of instruction in education through efficient and intelligent approaches, such as utilizing technologies like the WEKA tool for problem-solving. This is especially pertinent in the context of the changing educational environment, where Artificial Intelligence technologies are vital for increasing student learning experiences and overall educational advancement.

- ‘NN approaches’: specify what it is

Ans:

Neural Networks (NN) approaches.

- ‘FRs can effectively assess xxx’: add references

Ans:

Reference included.

- ‘An influence factors analysis model xxx’: explain the performance of such model

Ans:

The creation of an influence factors analysis model is frequently an iterative procedure, where the model is continuously improved as further data is obtained or as the comprehension of the phenomena grows more profound. The objective is to develop an effective tool for thoroughly analyzing the numerous aspects that contribute to the observed results.

- ‘teaching wushu’: what is wushu?

Ans:

Changed. 

- Add references for ‘Resources for teaching wushu xxx’

Ans:

Reference 10 utilized for this sentence.

6. Section 2 Related work has summarized too much literature, but not explaining why they are relevant to this work. If some don’t use the same method, I suggest removing them.

Ans: few literature survey is removed according to your comment. 

7. Figure 6: figure legend overlaps with some bars

Ans: Figure 6 changed accordingly.

Reviewer #3: This paper develops a powerful approach for assessing the influencing factors of college and university teaching effects by implementing the Sequential Intuitionistic Fuzzy (SIF) assisted Long Short-Term Memory (LSTM) model. The enhancement of the SIF-LSTM model for assessing the influencing factors of teaching quality is proved by the accuracy of 98.4%, Mean Square Error (MSE) of 0.028%, Tucker Lewis Index (TLI) measure for all influencing factors and entropy measure of non-membership and membership degree correlation of factors related to quality in teaching by various dimensional measures. The effectiveness of the proposed model is validated by implementing data sources with a set of 60+ teachers' and students' open-ended questionnaire surveys from a university.

However, there are some comments.

1- You must add qualitative comparison between your proposed scheme and some of existing methods based some metrices such as complexity time, methodology, preprocessing, and others.

Ans:

In addition, SIF-LSTM system efficiency is further evaluated using qualitative metric such as time investment for preparation and time efficiency for assessment. The obtained result is shown in Table 5 and the results are compared with existing methods such as AVGA-FBPNN, DL-ELSVM, CFNN, and DA-BP.

Table 5: Qualitative Analysis of the proposed SIF-LSTM

Parameter AVGA-FBPNN DL-ELSVM CFNN DA-BP Proposed SIF-LSTM

Time investment for preparation 78 hrs 72hrs 67hr 54hr 36hr

Time efficiency for assessment 64hr 60hr 58hr 50hr 35hr

From the table 5, it clearly states the proposed SIF-LSTM approach attains minimum preparation and assessment time for improving the teaching quality. The set of training samples and patterns are more useful to understand the concept that leads to minimize the time consumption. However, the system consumes high teaching quality and efficiency compared to other methods. 

2- In figure 3, the three boxes of membership degree, non-membership degree and hesitation degree do not connect any other box in the flowchart, I think there is something wrong.

Ans:

Three boxes are related to assessment qualities. According to these three factors, first and second assessment quality is checked and the obtained results are fed into the fuzzy scoring factor computations. 

3- the resolution of some figures needs some enhancements

Ans:

Figure resolutions are checked and changed accordingly.

---

## [Decision Letter · Decision Letter 1]

22 Mar 2024

PONE-D-23-38898R1Evaluation of Influencing Factors of China University Teaching Quality Based on Fuzzy Logic and Deep Learning TechnologyPLOS ONE

Dear Dr. Yu,

Thank you for submitting your manuscript to PLOS ONE. After careful consideration, we feel that it has merit but does not fully meet PLOS ONE’s publication criteria as it currently stands. Therefore, we invite you to submit a revised version of the manuscript that addresses the points raised during the review process.

We look forward to receiving your revised manuscript.

Kind regards,

Le Hoang Son, Ph.D

Academic Editor

PLOS ONE

Journal Requirements:

Reviewers' comments:

Reviewer's Responses to Questions

**Comments to the Author**

1. If the authors have adequately addressed your comments raised in a previous round of review and you feel that this manuscript is now acceptable for publication, you may indicate that here to bypass the “Comments to the Author” section, enter your conflict of interest statement in the “Confidential to Editor” section, and submit your "Accept" recommendation.

Reviewer #1: All comments have been addressed

Reviewer #2: All comments have been addressed

Reviewer #3: All comments have been addressed

2. Is the manuscript technically sound, and do the data support the conclusions?

Reviewer #1: Yes

Reviewer #2: Yes

Reviewer #3: Yes

3. Has the statistical analysis been performed appropriately and rigorously? 

Reviewer #1: Yes

Reviewer #2: Yes

Reviewer #3: Yes

4. Have the authors made all data underlying the findings in their manuscript fully available?

Reviewer #1: Yes

Reviewer #2: No

Reviewer #3: Yes

5. Is the manuscript presented in an intelligible fashion and written in standard English?

Reviewer #1: Yes

Reviewer #2: Yes

Reviewer #3: Yes

6. Review Comments to the Author

Reviewer #1: The whole article is properly written understandably. Moreover, this article sounds well with various aspects in this research area and the involvement of this work is appreciable.

Reviewer #2: (No Response)

Reviewer #3: Thanks all done. the authors have adequately addressed your comments raised in a previous round of review and you feel that this manuscript is now acceptable for publication..

7. PLOS authors have the option to publish the peer review history of their article (what does this mean?). If published, this will include your full peer review and any attached files.

Reviewer #1: **Yes: **Dr B Santhosh Kumar

Reviewer #2: No

Reviewer #3: **Yes: **Ahmed A. A. Gad-Elrab

---

## [Author Response · Author response to Decision Letter 1]

27 Mar 2024

Journal Requirements:

Thank you for your comment. I have replaced the retracted articles in the references section. Please check for further review.

Review Comments to the Author

Reviewer #1: The whole article is properly written understandably. Moreover, this article sounds well with various aspects in this research area and the involvement of this work is appreciable.

Thank you for your comment.

Reviewer #2: (No Response)

Reviewer #3: Thanks all done. the authors have adequately addressed your comments raised in a previous round of review and you feel that this manuscript is now acceptable for publication.

Thank you for your comment.

---

## [Decision Letter · Decision Letter 2]

29 Apr 2024

Evaluation of Influencing Factors of China University Teaching Quality Based on Fuzzy Logic and Deep Learning Technology

PONE-D-23-38898R2

Dear Dr. Yu,

We’re pleased to inform you that your manuscript has been judged scientifically suitable for publication and will be formally accepted for publication once it meets all outstanding technical requirements.

Kind regards,

Le Hoang Son, Ph.D

Academic Editor

PLOS ONE

Additional Editor Comments (optional):

Reviewers' comments:

Reviewer's Responses to Questions

**Comments to the Author**

1. If the authors have adequately addressed your comments raised in a previous round of review and you feel that this manuscript is now acceptable for publication, you may indicate that here to bypass the “Comments to the Author” section, enter your conflict of interest statement in the “Confidential to Editor” section, and submit your "Accept" recommendation.

Reviewer #2: All comments have been addressed

Reviewer #3: All comments have been addressed

2. Is the manuscript technically sound, and do the data support the conclusions?

Reviewer #2: Yes

Reviewer #3: Yes

3. Has the statistical analysis been performed appropriately and rigorously? 

Reviewer #2: Yes

Reviewer #3: Yes

4. Have the authors made all data underlying the findings in their manuscript fully available?

Reviewer #2: No

Reviewer #3: Yes

5. Is the manuscript presented in an intelligible fashion and written in standard English?

Reviewer #2: Yes

Reviewer #3: Yes

6. Review Comments to the Author

Reviewer #2: (No Response)

Reviewer #3: Thanks all done. the authors have adequately addressed your comments raised in a previous round of review and you feel that this لاmanuscript is now acceptable for publication.

---

## [Editor Report · Acceptance letter]

9 Jul 2024

PONE-D-23-38898R2 

PLOS ONE

Dear Dr. Yu, 

I'm pleased to inform you that your manuscript has been deemed suitable for publication in PLOS ONE. Congratulations! Your manuscript is now being handed over to our production team.

Kind regards, 

on behalf of

Prof. Le Hoang Son 

Academic Editor

PLOS ONE